Single-particle analysis of small extracellular vesicles from human follicular fluid unveils immunomodulatory PD-L1+ subpopulations and potentially fertility biomarkers

Bortot Barbara 1
Di Florio Roberta 1
Zito Gabriella 1
Valle Francesco 2
Brucale Marco 2
Ricci Giuseppe 1 3
Vigano Paola 4
Biffi Stefania stefania.biffi@burlo.trieste.it 1
1 Obstetrics and Gynaecology Department, IRCCS Materno Infantile Burlo Garofolo , Trieste , Italy
2 Institute of Nanostructured Materials, Istituto per lo Studio dei Materiali Nanostrutturati, Consiglio Nazionale delle Ricerche, Unitá Operative di Supporto di Bologna , Bologna , Italy
3 Department of Medical, Surgical and Health Science, University of Trieste , Trieste , Italy
4 Infertility Unit, Fondazione IRCCS Ca’ Granda Ospedale Maggiore Policlinico , Milan , Italy
Uversky Vladimir
Electronic publication date: 2025 Oct 28
Publication date: 2025
Volume: 13
Electronic Location ID: e20057
Received 2025 May 15; Accepted 2025 Aug 17
Copyright: ©2025 Bortot et al.
Copyright year: 2025
Copyright holder: Bortot et al.
License: This is an open access article distributed under the terms of the Creative Commons Attribution License, which permits unrestricted use, distribution, reproduction and adaptation in any medium and for any purpose provided that it is properly attributed. For attribution, the original author(s), title, publication source (PeerJ) and either DOI or URL of the article must be cited.
License URL: https://creativecommons.org/licenses/by/4.0/

Keywords: Atomic force microscopy, Follicular fluid, PD-L1, Small extracellular vesicles, Single-particle analysis

Funding: The Italian Ministry of Health, throught the contribution given to the Institute for Maternal and Child Health IRCCS Burlo Garofolo, Trieste –Italy This work was supported by the Italian Ministry of Health, throught the contribution given to the Institute for Maternal and Child Health IRCCS Burlo Garofolo, Trieste –Italy. The funders had no role in study design, data collection and analysis, decision to publish, or preparation of the manuscript.

==============================
In certain cell systems, small extracellular vesicles bearing PD-L1 (PD-L1+ sEVs) have been shown to suppress T-cell immunity. We investigated whether a distinct profile of PD-L1+ sEVs exists in human follicular fluid (FF), a microenvironment where immune tolerance is crucial for proper follicular development. We characterized the expression and colocalization of CD63, CD81, CD9, and PD-L1 in sEVs derived from FF of women undergoing fertility treatments (n = 10), utilizing single-particle interferometric reflectance imaging sensing combined with single-particle antibody capture and immunofluorescence labeling. Additionally, sEV size distribution was analysed via atomic force microscopy. These integrated techniques revealed that the majority of tetraspanin-expressing EVs in human FF are smaller than 50 nm. Statistical analysis revealed a significant difference in PD-L1 co-expression across CD63, CD81, and CD9, confirming a preferential association of PD-L1 with CD9+ sEVs. Coefficients of variation across the cohort further indicated that PD-L1/CD9 co-expression was the most consistent among patients, suggesting a stable and distinct sEV subpopulation. These findings underscore the potential of PD-L1+ sEVs as biomarkers for immune regulation in reproductive treatments. The discovery of distinct PD-L1+ sEV subpopulations suggests a role in modulating immune responses within the follicular microenvironment. Further studies are warranted to investigate the functional relevance of these vesicles in predicting fertility outcome, promoting local immune tolerance, and facilitating follicular development.

Introduction

Portions of this text were previously published as part of a preprint (https://doi.org/10.1101/2024.12.17.628903).

The follicular fluid (FF) surrounds developing oocytes in the ovaries, and its composition can significantly affect the success of fertilization and implantation (Revelli et al., 2009; Dumesic et al., 2015). FF is composed of active substances derived from the blood and secreted by granulosa and theca cells (Pan, Pan & Zhang, 2024). It contains a variety of hormones, proteins, amino acids, enzymes, fatty acids, cytokines, blood-clotting factors, and other molecules that play essential roles in supporting follicle maturation (Da Broi et al., 2018; Saint-Dizier et al., 2019). Changes in FF composition reflect variations in the secretion of granulosa and theca cells, the inner membrane, and blood plasma, influenced by both normal and abnormal physiological processes (Edwards, 1974).

Extracellular vesicles (EVs) in FF have a distinctive composition and carry cargoes of messenger RNAs (mRNAs), miRNAs, proteins, and metabolites (Théry et al., 2018). These particles are actively released into the extracellular space and play an important role in cell-to-cell communication by delivering their cargo to target cells (Tkach & Théry, 2016; Salomon et al., 2022; Muraoka et al., 2024). Several studies have focused on the functional significance of EVs collected in FF in promoting cell-to-cell communication during the reproductive process (Tesfaye et al., 2020; Wang et al., 2021; Salehi et al., 2023). For example, Wang et al. (2021) studied noncoding RNAs present in small EVs (sEVs) in FF as potential pregnancy predictors, offering new avenues for utilizing EVs in assisted reproduction. In another study, the increase in the number of small EVs in the FF from older women has been correlated to an enhanced release from somatic follicular cells, likely controlled by TP53 signaling pathways (Battaglia et al., 2020). Additionally, six miRNAs were discovered to be upregulated in the follicular fluid of older women (Battaglia et al., 2020).

Recent research has shown that immune cells residing in the ovaries and fallopian tubes, such as T cells and macrophages, as well as nonimmune cells like granulosa cells and oocytes, express PD-1 and its ligands PD-L1 and PD-L2 (Johnson et al., 2019; Johnson et al., 2023). PD-1 and its soluble ligands have been found in high concentrations in sEV fractions in human FF (Luu et al., 2020), at levels capable of controlling T-cell PD-1 activation when taken up, suggesting a regulatory role in immune responses (Johnson et al., 2023). Additionally, studies have shown that PD-L1 on EVs can promote tumor immune evasion by inducing T-cell exhaustion and inhibiting T-cell proliferation, highlighting PD-L1′s role in immune escape mechanisms employed by cancer cells (Chen et al., 2018). Therefore, activation of the PD-1 checkpoint pathway may be crucial for normal ovarian activities, such as follicle development and the establishment of immunological tolerance for germline cells and embryos (Johnson et al., 2023). It is noteworthy that the therapeutic potential of EVs expressing PD-L1 has been demonstrated in the context of primary ovarian insufficiency (POI), a clinical disorder characterized by the premature depletion of primordial follicles, resulting in compromised ovarian function (Touraine et al., 2024). Immune dysregulation plays a central role in the pathogenesis of POI, primarily by enhancing T cell activity within the ovarian microenvironment. A study in a murine model of POI showed that genetically engineered EVs overexpressing PD-L1 and Galectin-9 (Gal-9) effectively suppressed T cell infiltration and preserved ovarian function (Gu et al., 2025). These EVs promoted T cell apoptosis and mitigated disease progression, underscoring their therapeutic potential in the management of immune-mediated ovarian dysfunction (Gu et al., 2025).

Despite the recognized importance of sEVs in reproductive biology, current understanding of their heterogeneity and functional subpopulations within human FF remains limited. In particular, the presence, phenotypic characteristics, and potential immunoregulatory functions of PD-L1+ sEV subsets have not been systematically examined using high-resolution single-particle analysis. Moreover, many sEVs in FF fall below 50 nm in diameter, and conventional techniques such as flow cytometry and nanoparticle tracking analysis (NTA) exhibit limited sensitivity and resolution at this scale (Bachurski et al., 2019). To overcome these limitations, we applied single-particle interferometric reflectance imaging sensing (SP-IRIS) to analyze individual EVs captured via specific antibodies. SP-IRIS also supports multiplexed profiling of tetraspanin surface markers, including CD9, CD63, and CD81, and their co-expression patterns (Théry et al., 2018). Tetraspanins serve not only as canonical EV markers, but also play integral roles in EV biogenesis, molecular cargo sorting, and intercellular communication, making their profiling essential for elucidating EV functionality (Andreu & Yáñez Mó, 2014). When combined with atomic force microscopy (AFM), SP-IRIS enables precise morphological and dimensional characterization of vesicles smaller than 50 nm, a critical advantage given the size distribution observed in this study.

In order to investigate the intrafollicular communication mechanisms that regulate T lymphocyte immunity, the purpose of this study was to assess whether a distinct profile of PD-L1+ sEVs exists in human FF. This aim has relevance in the context of the recent demonstration for a role of PD-L1+ sEVs beyond tumor immunity (Yu, Liu & Chen, 2022).

Materials and Methods

Patient recruitment

Ten patients who underwent assisted reproductive technology (ART) treatment were enrolled in this study (Table 1). Following approval by the Institutional Review Board of the Institute for Maternal and Child Health IRCCS Burlo Garofolo, Trieste, Italy (IRB-BURLO 07/2023 dd. 31.08.2023), patients were asked to sign an informed consent form. The FF from the first and largest punctured follicle from the bilateral ovaries was collected during the oocyte retrieval procedure (transvaginal follicular aspiration). Each ovarian follicle was aspirated independently.

Table 1 Patient characteristics.

Pt	Age	E2*
pg/ml	P4*
ng/ml	Final trigger	N oocytes	Endometriosis	Pregnancy	Live birth	
1	33	324	2	UP- hCG 10000 UI (Gonasi 10000®)	24		Yes**	Yes	
2	34	2,945	0,75	UP- hCG 3300 UI (Gonasi 3300®)	9		Yes **	Yes	
3	35	1,203	0,6	UP- hCG 10000 UI (Gonasi 10000®)	4	Yes	No		
4	41	683	0,7	r-hCG 6500 UI (Ovitrelle250®)	2		No		
5	40	1,371	1	r- hCG 6500 UI (Ovitrelle250®)	2		No		
6 #	33	572		0.2 mg triptorelin (Decapeptyl® )	20		No		
7	43	2,288	1,3	r-hCG 6500 UI (Ovitrelle250®)	3		No		
8	31	2,000	0,9	0.2 mg triptorelin (Decapeptyl® )	6		Yes**	Yes	
9	35	2,574	1,7	HP-hCG 10000 UI (Gonasi 10000®)	16	Yes	Yes**	No	
10	39	3,172	08	r-hCG 6500 UI (Ovitrelle250®)	7		No **		
Notes.

# Fertility preservation in breast cancer BRCA neg.

* Serum level on the day of trigger.

** Frozen embryo transfer.

Abbreviations Pt patient

E2 oestradiol

P4 progesterone

UP-hCG urine pregnancy human chorionic gonadotropin

r-hCG recombinant human chorionic gonadotropin

HP-hCG highly purified human chorionic gonadotropin

BRCA Breast cancer gene

Inclusion criteria: Follicular fluid samples were collected from infertile patients undergoing in vitro fertilization. Exclusion criteria: To ensure sample purity, the presence of blood contamination was assessed by visual inspection. Samples that appeared cloudy or blood-stained were excluded from the study. Only uncontaminated samples were processed for analysis.

Single-particle interferometric reflectance imaging sensing: ExoView R100 analysis

The investigation was performed via single-particle interferometric reflectance imaging sensing (SP-IRIS) via the ExoView R100 system, which enables the detection of particles as small as 50 nm through optical interferometry on Si/SiO2 substrates integrated into the microarray chip (Mizenko et al., 2021). Initially, 35 µL of each sample, diluted 1:10, were applied to the microarray chip functionalized with capture antibodies specific for CD9, CD81, and CD63 (ExoView™ Tetraspanin Kit, CD9, CD63, CD81 + IgG Negative Control (mIgG); Nano View Bioscience, Boston, MA, USA) and incubated for 16 h at room temperature. The chips were washed three times with solution A from the ExoView Human Tetraspanin Kit. Each binding event is imaged using visible light interference to determine particle size and concentration. Following EV capture, fluorescently labeled antibodies, anti-CD9 CF488, anti-CD81 CF555 (from the ExoView™ Tetraspanin Kit), and anti-PD-L1 [28-8] Alexa Fluor® 647 (ab209960, Abcam, Cambridge, UK), were applied to detect surface markers. The antibodies were first diluted 1:500 in Solution A (Nano View Bioscience), and then mixed 1:1 with fresh Solution A, achieving a final dilution of 1:1000. Chips were incubated with this antibody mix for 60 min at ambient temperature, then washed and dehydrated. Image acquisition was performed using the ExoView R100 reader and ExoView Scanner 3.0 software. Data were subsequently analyzed using ExoView Analyzer 3.0.

To ensure the reliability and specificity of the immunofluorescence labelling results, the ExoView R100 system integrates crucial internal controls. Fluorescence specificity and intensity cut-offs were established using isotype control mouse IgG (mIgG) in accordance with the manufacturer’s guidelines. These isotype-matched negative controls assess non-specific antibody binding, ensuring that detected signals specifically originate from tetraspanins and PD-L1 expressed on sEVs. Furthermore, the system’s software performs real-time quality checks during data acquisition, automatically flagging samples with elevated background levels. This automated detection is vital for identifying and mitigating technical artifacts such as inadequate washing, particle aggregation, or sample/reagent impurities, thereby enabling researchers to exclude unreliable data or optimize experimental conditions. Anomalous spots were excluded to reduce analytical variability. Co-localization analysis was also performed to determine marker overlap across individual particles (Breitwieser et al., 2022).

Size exclusion chromatography isolation on qEVoriginal 35 nm columns, IZON

sEVs were isolated via size exclusion chromatography (SEC) following the manufacturer’s protocol. Briefly, single qEV original 35 nm columns (Izon Science; distributed by Shaefer-Tec, Italy) were brought to room temperature and equilibrated with 17 mL of freshly filtered (0.22 µm) phosphate-buffered saline (PBS 1X, Corning, Glendale, AZ, USA). FF samples were initially centrifuged at 2,000 × g for 30 min. Subsequently, 500 µL of the FF supernatant was carefully loaded onto the top of the column. Once the sample had fully entered the resin bed, 0.22-µm-filtered PBS 1X was added to the column to initiate elution of EVs. Fractions were manually collected as follows: the void volume (F0) was 1.6 mL, followed by six subsequent fractions (F1 to F6), each consisting of 400 µL. Fractions F1–F6, which are expected to contain sEVs, were pooled and concentrated via ultracentrifugation at 100,000 × g for 70 min. Following SEC, samples were frozen at −80 °C immediately. The fractions were analyzed immediately after thawing, ensuring minimal degradation and aggregation prior to the AFM analysis.

Atomic force microscopy morphometry

Samples obtained via SEC isolation were analyzed via atomic force microscopy (AFM) as described elsewhere (Ridolfi et al., 2020; Ridolfi et al., 2023; Bortot et al., 2022). Briefly, images were taken in PeakForce mode on a Bruker Multimode8 equipped with a Nanoscope V controller, a sealed fluid cell and a type JV piezoelectric scanner using Bruker ScanAsystFluid+ probes (triangular cantilever, nominal tip curvature radius 2–12 nm, nominal elastic constant 0.7 N/m) calibrated with the thermal noise method. Image analysis was performed with a combination of Gwyddion 2.58 (Nečas & Klapetek, 2012) and custom Python scripts to recover the equivalent spherical diameter of individual particles. All AFM images were acquired at combinations of scan size and pixel size resulting in a fixed resolution of 9.77 nm/pixel. Scanning speed was kept below five µm/s. Particles were selected as described in (Ridolfi et al., 2020; Ridolfi et al., 2023); briefly, a height threshold of eight nm was applied to single out all particles from the background, then putative vesicles were selected for subsequent morphometrical analysis by removing particles that (i) touched any edge of the image, (ii) had a maximum inscribed disc radius below eight nm, (iii) had a maximum value below 10 nm. At least 100 individual particles were measured for each sample.

Statistical analysis

All statistical analyses were performed using Python, leveraging the scipy.stats and statsmodels libraries. Graphs were performed using GraphPad Prism 10.5.0. Data are presented as mean ± standard deviation (SD) or as percentages, as indicated in the figure legends and text.

For comparisons of particle counts between total particles and those detected via imaging mode (as illustrated in Figs. 1 and 2), paired t-tests were applied to triplicate analyses for individual samples and across the entire cohort, respectively. To assess differences in total particle numbers captured by different tetraspanin antibodies (CD63, CD81, CD9), as shown in Fig. 3, paired t-tests were utilized for direct comparisons between specific capture antibody groups. For the analysis of PD-L1 co-expression ratios across CD63, CD81, and CD9 (as presented in the relevant results section), a repeated measures Analysis of Variance (ANOVA) was conducted to determine if there was an overall statistically significant difference among the mean values of these ratios. Following a significant ANOVA result, paired-samples t-tests were performed as post-hoc analyses to investigate specific pairwise differences between the ratios. Coefficients of variation (CV) were calculated for PD-L1 co-expression ratios across the 10-patient cohort to assess inter-individual variability and consistency of sEV subpopulations. The CV is a standardized measure of relative variability, calculated as the ratio of the standard deviation to the mean, expressed as a percentage. A lower CV indicates greater consistency within the data set. A significance level of α = 0.05 was set for all statistical tests. Specific p-values are reported in the Results section.

Figure 1 Characterization of follicular fluid small extracellular vesicles (sEVs) using the ExoView 100 Platform.

This figure illustrates the expression profiles and co-localization patterns of tetraspanin markers on individual sEVs isolated from human follicular fluid. The graph presents a representative dataset from a single patient sample, highlighting tetraspanin co-localization. sEVs were captured on distinct spots coated with anti-CD63, anti-CD81, and anti-CD9 antibodies. Detection was performed via fluorescent labeling using anti-CD81, anti-CD9, and anti-PD-L1 antibodies. EV populations were visualized using ExoView fluorescence mode, which detects particles as small as 35 nm. Single-particle interferometric reflectance imaging (IM) quantified vesicles bound to capture spots, typically ranging from 50–200 nm. Bars represent mean values from three technical replicates, with standard deviations indicated. Asterisks (****) indicate a p-value less than 0.0001 (p < 0.0001).

Figure 2 Quantification of small extracellular vesicles (sEVs) captured by tetraspanin antibodies: comparison of total vs. IM-detected particle counts across patient cohort.

The panel presents cohort data, comparing total captured particle counts with those detected exclusively by IM mode for each tetraspanin capture antibody (CD63, CD81, CD9) across the full patient cohort. Bars represent mean values, with standard deviations reported alongside individual cohort values (indicated by dots). Asterisks (****) indicate a p-value less than 0.0001 (p < 0.0001). IM, Single-particle interferometric reflectance imaging.

Figure 3 Cohort-wide analysis of small extracellular vesicles (sEVs) capture abundance and distribution across tetraspanin antibody spots.

The panel details the cohort data on overall captured particle abundance and distribution across tetraspanin capture spots. The bar chart displays average total particle counts on the CD9, CD63, and CD81 antibody capture spots from the full cohort, while the embedded pie chart illustrates the mean fractional distribution of captured particles across these three tetraspanin antibody capture spots from the full cohort. Asterisks (**) indicate a p-value less than 0.01 (p < 0.01).

Results

ExoView R100 analysis reveals the presence of sEVs with a distinct pattern of PD-L1 and tetraspanin profiles in FF

A comprehensive characterization of the tetraspanin expression profile in FF sEVs has not been accomplished; hence, we performed this investigation using a platform that minimized biases derived from the sEV isolation step and was capable of characterizing vesicles smaller than 50 nm (Mizenko et al., 2021). We have associated the analysis of tetraspanins with that of our target marker, PD-L1. The ExoView R100 platform was used to quantify distinct EV populations by combining two complementary systems: antigenic capture via interferometric imaging and fluorescent antigenic detection of the captured EV. Single-particle interferometric reflectance imaging (IM) enables the quantification of particles bound to the capture spot between 50 and 200 nm (Avci et al., 2015), whereas fluorescence mode was implemented to quantify the particles as small as 35 nm. The concurrent use of three distinct fluorescence-labelled antibodies enabled the assessment of antigen distribution on a single sEV (Fig. 1).

As shown in Fig. 1, a representative example of the EXOVIEW 100 output from one patient sample, the number of particles detectable in imaging mode was significantly lower than the total number of particles detected in fluorescence mode, consistent with the fact that particles larger than 50 nm accounted for less than 10% of the total population. A paired t-test was applied to triplicate analysis comparing total particles and those detected via imaging mode, revealing statistically significant differences. Specifically, in this representative example, CD63 yielded p = 1.40  × 10−6 (imaging mode: 9%), CD81 p = 9.92  × 10−7 (imaging mode: 5%), and CD9 p = 1.81  × 10−5 (imaging mode: 7%). Furthermore, analysis of the entire cohort (Fig. 2) consistently demonstrated that the majority of sEVs were smaller than 50 nm. Specifically, particles under 50 nm constituted 10.38% of the total for CD63-captured EVs (p = 5.60  × 10−7), 5.15% for CD81-captured EVs (p = 3.02  × 10−9), and 8.86% for CD9-captured EVs (p = 5,3  × 10−8).

Figure 3 illustrates the overall abundance and distribution of sEVs captured on different tetraspanin antibodies from the full patient cohort. The bar chart, titled “Total particle number,” displays the average particle counts on the CD63, CD81, and CD9 antibody capture spots. Each bar represents the mean particle count, with individual data points (dots) showing the counts from each patient in the cohort, and error bars indicating the standard deviation. Statistical analysis reveals a significant difference in the total number of captured particles: specifically, the number of particles captured by anti-CD9 antibodies is significantly lower than those captured by anti-CD81 antibodies (p = 0.0059) and anti-CD63 antibodies (p = 0.0047).

Figure 4 illustrates the multi-marker co-expression profiles of sEVs within the follicular fluid, derived from the full patient cohort. It provides a detailed breakdown of the percentage of sEVs exhibiting various combinations of tetraspanins (CD63, CD81, CD9) and PD-L1, relative to their respective capture antibodies. A small but detectable proportion of sEVs exhibited PD-L1 expression on the CD9, CD63, and CD81 antibody capture spots. This highlights the presence of PD-L1-bearing sEVs within the follicular microenvironment, a finding of particular interest given their potential immunomodulatory roles. While PD-L1 expression is observed, its overall fraction remains lower compared to the robust tetraspanin co-localization. The detailed breakdown across various capture and detection combinations provides nuanced insights into the complexity of the sEV proteome in follicular fluid.

Figure 4 Cohort-wide analysis of multi-marker co-expression profiles in follicular fluid small extracellular vesicles (sEVs).

The figure provides cohort data on the multi-marker co-expression profiles of follicular fluid sEVs. This protein co-localization analysis was performed based on single-EV protein expression data, with percentages representing the total fluorescence counts corresponding to each color, indicating the specific co-expression combination of each protein across the full cohort.

Characterization of PD-L1 expression profile of sEVs in FF

The analysis of the PD-L1 expression profile revealed a preference for colocalization with CD9+ sEVs (Fig. 5). In terms of the three capture spots, CD63, CD81, and CD9, the percentages of EVs expressing PD-L1 were 2,43%, 2,62% , and 4,77% respectively (Fig. 5). A repeated measures analysis of variance (ANOVA) was conducted to assess if there was an overall statistically significant difference among the mean values of these three PD-L1 ratios. The ANOVA revealed a statistically significant overall difference among the markers (F(2,18) = 5.71, p = 0.0120). To further investigate these differences, paired-samples t-tests were subsequently performed as post-hoc analyses. The findings indicate that while the PD-L1/CD63 and PD-L1/CD81 ratios do not significantly differ from each other (t(9)= −0.51, p = 1.00), both are significantly different from the PD-L1/CD9 ratio. Specifically, a statistically significant difference was observed for PD-L1/CD63 vs. PD-L1/CD9 (t(9)= −4.99, p = 0.0024). Similarly, a statistically significant difference was also found for PD-L1/CD81 vs. PD-L1/CD9 (t(9)= −4.45, p = 0.0045) (Fig. 6). These results confirm a preferential association of PD-L1 with CD9+ sEVs.

Figure 5 PD-L1 expression profiles and co-expression patterns on individual small extracellular vesicles (sEVs).

Protein colocalization analysis of the follicular fluid samples on the CD9, CD63, and CD81 antibody capture spots. Pie charts illustrate the percentages of PD-L1+ sEV subpopulations with different tetraspanin combinations.

Figure 6 PD-L1 expression in small extracellular vesicles (sEVs) captured via CD9, CD63, and CD81: co-localization analysis and statistical significance.

Percentage of sEVs expressing PD-L1 in combination with single or multiple tetraspanins. Bars represent mean percentages of PD-L1+ EVs captured on each tetraspanin-specific spot. Error bars indicate standard deviation. An asterisk (*) indicates a p-value less than 0.01 (p < 0.01).

We observed that PD-L1 mostly colocalized with a single tetraspanin, with percentages of 1.88%, 1.95% and 3.98%, respectively, while the percentage of EVs exhibiting PD-L1 in combination with multiple tetraspanins was less than 1% (Fig. 6). Coefficients of variation (CV) across the 10-patient cohort were 0.64 for PD-L1/CD63, 0.62 for PD-L1/CD81, and 0.18 for PD-L1/CD9, indicating that PD-L1/CD9 co-expression was the most consistent among patients. This lower CV suggests reduced inter-individual variability and a more stable association between PD-L1 and CD9+ sEVs, potentially reflecting a conserved biological subpopulation. In contrast, the higher CVs for PD-L1/CD63 and PD-L1/CD81 point to greater heterogeneity, which may be influenced by patient-specific factors or follicular microenvironmental conditions. These findings suggest that PD-L1+ sEVs are predominantly associated with individual tetraspanin subpopulations, particularly CD9+ vesicles, and that multi-marker PD-L1+ sEVs are rare and potentially more heterogeneous.

It has recently been demonstrated that EVs exhibit a distinct nanomechanical fingerprint that can be identified using force spectroscopy. This fingerprint can be used to distinguish between various EV populations, making atomic force microscopy (AFM)-based nanomechanics a useful tool for evaluating EV identity, purity, and function (Bortot et al., 2021; Ridolfi et al., 2023). Single-particle AFM morphometry performed on SEC-enriched sEV samples revealed that over 50% of the particles had diameters smaller than 50 nm, while no significant amounts of vesicles with diameters above 100 nm were observed (Fig. 7).

Figure 7 Atomic force microscopy (AFM) morphometry.

Representative AFM micrographs of three sEV samples from the cohort (left) and corresponding size distributions (right) obtained via quantitative morphometry on at least five 5 × 5 µm AFM images per sample. The number of analyzed particles were N = 439, N = 119 and N = 168 for the three samples. Diameters obtained via quantitative morphometry of the particles are displayed as box plots (right) in which the central horizontal lines correspond to the median value and the small cross to the arithmetic mean of each distribution. Boxes comprise values from the 25th to the 75th size percentile, while whiskers extend to values within 1.5 times the interquartile ranges. Outlier points are represented as individual points.

Discussion

EVs have emerged as important mediators and regulate several reproductive processes, including gamete development and implantation (Giacomini et al., 2017; Smith & Russell, 2022; Makieva et al., 2024). Embryo-derived EVs and placental EVs carry immunomodulatory molecules such as HLA-G, progesterone-induced blocking factor, and regulatory miRNAs, which can influence maternal immune cell behavior and promote tolerance (Giacomini et al., 2017; Rosenfeld, 2024). A growing body of evidence supports the idea that FF sEVs ensure the successful development of follicles and oocytes (Tesfaye et al., 2020; Wang et al., 2021; Gonzalez Fernandez et al., 2023). Furthermore, adverse pregnancy outcomes in ART have been linked to the low quality of embryos (Fauque et al., 2007), with FF sEVs being potential biomarkers for predicting pregnancy (Muraoka et al., 2024). However, EV characterization in human FF is still in its early stages, and more research in this area is needed to fully understand the potential of EVs in FF and their impact on female reproductive biology (Giacomini et al., 2020).

Recent findings suggest a potential immune-regulatory role for PD-L1+ sEVs within the ovarian microenvironment (Johnson et al., 2019; Johnson et al., 2023; Gu et al., 2025), with potential implications for ART outcomes. At the maternal-fetal interface, PD-L1 expression is essential for establishing immune tolerance, and its dysregulation has been linked to pregnancy complications such as preeclampsia and recurrent miscarriage (Meggyes et al., 2019). Similarly, in ovarian pathologies like polycystic ovary syndrome (PCOS), PD-L1 expression is reduced, contributing to granulosa cell apoptosis and chronic inflammation via the PI3K/AKT pathway (Han et al., 2025). These immune imbalances may impair follicular development and oocyte quality, which are critical determinants of ART success.

Trophoblasts also express PD-L1, which interacts with PD-1 on maternal immune cells to induce M2 macrophage polarization and dampen proinflammatory cytokine production; blockade of PD-L1 in animal models has been shown to increase fetal loss, underscoring its protective role (Zhang, Liu & Sun, 2023). Similarly, PD-L1 expression on syncytiotrophoblasts in third-trimester placentas and retained products of conception suggests a mechanism by which the fetus evades maternal immune rejection (Singh, 2023). The presence of PD-L1+ EVs in reproductive fluids may represent a novel mechanism of immune regulation akin to that observed in placental EVs. However, while our data suggest a potential immune-regulatory role for PD-L1+ sEVs in the ovarian microenvironment, these implications remain speculative in the absence of functional validation. Acknowledging this limitation, future studies should incorporate targeted assays to assess the immunomodulatory effects of PD-L1+ sEVs, which would be essential to substantiate their role and clarify their relevance within the broader context of reproductive immunology.

On the basis of our data, most tetraspanin-expressing EVs in human FF are smaller than 50 nm. This contrasts with previous studies on human samples. Several studies have successfully isolated FF sEVs via serial centrifugation or precipitation and revealed that sEVs are less than 200 nm in diameter, with a mean size of approximately 100 nm (Zhou et al., 2024; Gu et al., 2024; Muraoka et al., 2024). The different techniques used for EV analysis may account for this difference. The strengths of the present findings are that we followed an innovative approach to identify and characterize EVs from unpurified biological sources. The Exoview R100 platform allows direct capture of EVs from biofluids via tetraspanin-specific antibodies, eliminating the need for prior separation. Moreover, SP-IRIS on the ExoView R100 platform enables the detection of EVs smaller than 50 nm, a capability that other methods such as nanoparticle tracking analysis (NTA) cannot achieve (Bachurski et al., 2019). AFM analysis of sEVs at the single-vesicle level revealed their size distribution, confirming that most sEVs are under 50 nm in size (Ridolfi et al., 2020; Ridolfi et al., 2023).

The tetraspanins CD9, CD63, and CD81 are widely recognized as the most prevalent markers associated with EVs in the scientific literature (Karimi et al., 2022; Okada-Tsuchioka et al., 2022). These markers have been extensively used in various research investigations, such as ELISA, flow cytometry, and lab-on-a-chip assays, to capture EVs comprehensively. We observed consistent patterns in the EV subpopulations based on tetraspanins, with anti-CD81+ sEVs representing the majority of sEVs in all samples and antibody-capture spots. In particular, 42%, 46%, and 50% of all the particles captured by anti-CD63, anti-CD81, and anti-CD9 antibodies, respectively, were positive for CD81. These results are consistent with prior studies linking CD81 and CD9 in the context of reproductive biology (Takahashi et al., 2001; Kaji et al., 2002). In particular, Rubinstein et al. (2006) studied the fertility of mice lacking the CD9 and CD81 genes, suggesting that CD9 and CD81 are crucial for successful sperm-egg interactions.

CD9+ sEVs have the highest PD-L1 expression at 5%. Additionally, we observed that PD-L1 was predominantly associated with a single tetraspanin protein rather than multiple tetraspanins in sEVs. The tetraspanin CD9 is expressed by all the major leukocyte subsets, including B cells, CD4+ T cells, CD8+ T cells, natural killer cells, granulocytes, monocytes, macrophages, and both immature and mature dendritic cells, as well as at elevated levels by endothelial cells (Reyes et al., 2018). Notably, CD9 mediates the interaction between sperm and eggs during fertilization (Le Naour et al., 2000; Miyado et al., 2000; Kaji et al., 2000; Rubinstein et al., 2006; Chalbi et al., 2014), allowing their membranes to fuse for successful fertilization. The fact that PD-L1 expression is much greater in the CD9+ sEV population, which is actually less abundant than the CD63+ and CD81+ subsets, suggests that CD9 might play a particular role in delivering PD-L1. A weakness of the present study relates to the limited number of FF samples analysed. On the other hand, results were consistent among samples as demonstrated by the narrow range of variability in the data.

In conclusion, the presence of PD-L1 in FF sEVs indicates potential roles in modulating immune signaling within the ovarian microenvironment, warranting further investigation through functional studies. Although sample size does not allow for correlation analysis between protein expression on sEVs and clinical parameters or pregnancy outcomes, these findings highlight the importance of further research into the role of the PD-1 checkpoint pathway in female reproduction.

Supplemental Information

Supplemental Information 1 Graphical Abstract

PD-L1+ small extracellular vesicles (sEVs) in human follicular fluid were analyzed to explore their role in immune regulation during follicular development. Using single-particle interferometric reflectance imaging sensing with antibody capture and immunofluorescence labeling, along with atomic force microscopy, we characterized sEVs from FF of women undergoing fertility treatment. Most tetraspanin+ sEVs were smaller than 50 nm. PD-L1 showed preferential co-expression with CD9+ sEVs, with significantly higher association compared to CD63 or CD81.

Abbreviations

AFM Atomic force microscopy

ART assisted reproductive technology ART

BRCA Breast cancer gene

E2 oestradiol

FF follicular fluid

HP-hCG highly purified human chorionic gonadotropin

P4 progesterone

PD-L1 Programmed Death-Ligand 1

PD-L2 Programmed Death-Ligand 2

Pt patient

r-hCG recombinant human chorionic gonadotropin

SEC size exclusion chromatography

sEVs small extracellular vesicles

SP-IRIS single-particle interferometric reflectance imaging sensing

UP-hCG urine pregnancy human chorionic gonadotropin

Additional Information and Declarations

Competing Interests

Author Contributions

Human Ethics

Data Availability

The authors declare there are no competing interests.

Barbara Bortot conceived and designed the experiments, performed the experiments, analyzed the data, prepared figures and/or tables, authored or reviewed drafts of the article, and approved the final draft.

Roberta Di Florio performed the experiments, authored or reviewed drafts of the article, and approved the final draft.

Gabriella Zito conceived and designed the experiments, authored or reviewed drafts of the article, and approved the final draft.

Francesco Valle performed the experiments, analyzed the data, prepared figures and/or tables, authored or reviewed drafts of the article, and approved the final draft.

Marco Brucale performed the experiments, analyzed the data, prepared figures and/or tables, authored or reviewed drafts of the article, and approved the final draft.

Giuseppe Ricci conceived and designed the experiments, authored or reviewed drafts of the article, and approved the final draft.

Paola Vigano analyzed the data, authored or reviewed drafts of the article, and approved the final draft.

Stefania Biffi conceived and designed the experiments, performed the experiments, analyzed the data, prepared figures and/or tables, authored or reviewed drafts of the article, and approved the final draft.

The following information was supplied relating to ethical approvals (i.e., approving body and any reference numbers):

The ethics were approved by the Institutional Review Board of the Institute for Maternal and Child Health IRCCS Burlo Garofolo, Trieste, Italy (IRB-BURLO 07/2023 dd. 31.08.2023).

The following information was supplied regarding data availability:

The data is available at Zenodo: Biffi, S. (2025). Single-particle analysis of small extracellular vesicles from human follicular fluid unveils immunomodulatory PD-L1+ subpopulations and potentially fertility biomarkers [Data set]. Zenodo. https://doi.org/10.5281/zenodo.16094765.

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
