# Peer review of "Single-particle analysis of small extracellular vesicles from human follicular fluid unveils immunomodulatory PD-L1+ subpopulations and potentially fertility biomarkers"

_PeerJ, doi:10.7717/peerj.20057_

## Round 0.1 · original submission · Major Revisions

Reviewer 1 ·

Basic reporting

Title & Abstract

Title:
The title of the manuscript is “Single-particle analysis of small extracellular vesicles from the human follicular fluid reveals distinct PD-L1+ populations.”
The title might benefit by adding the following specificity about the functional significance of the findings or how they relate to broader implications. For example, the role of PD-L1+ sEVs in immune modulation or fertility outcomes could enhance the appeal to a wider audience.
Suggested Revised Title: Single-particle analysis of small extracellular vesicles from human follicular fluid unveils immunomodulatory PD-L1+ subpopulations and potential fertility biomarkers.
This modification emphasizes the broader implications and relevance of the findings.

Abstract:
The abstract explains the findings of the study well, but it does not highlight the novelty explicitly, like why this research stands out compared to prior studies, especially those on sEVs and reproduction. The functional significance of PD-L1+ populations is acknowledged as requiring further research, but could have been elaborated with a more specific hypothesis or potential clinical/biological implications. The phrasing of "further research is needed to fully understand functional significance" could come across as generic. Instead, a more active phrase outlining future directions would make the abstract more impactful.
Highlight the novelty of the study directly, such as the innovative use of SP-IRIS or the importance of profiling tetraspanins at the single-vesicle level.
1. Include a sentence on the broader implications for reproductive immunology or clinical applications, focusing on precision fertility treatments. For instance, “These findings underscore the potential of PD-L1+ sEVs as non-invasive biomarkers for immune regulation in reproductive treatments.”
2. Rephrase the last sentence to suggest specific future research, such as exploring the potential of PD-L1+ sEVs in predicting fertility outcomes.

Introduction
The introduction section of the manuscript provides a solid foundation for explaining the context and key elements. It introduces important concepts such as follicular fluid, small extracellular vesicles, tetraspanins, and the immune-related PD-1/PD-L1 axis. However, there are areas where the background could be improved by adding more depth and clarity to create a stronger narrative and provide a more comprehensive understanding of the significance of the study.

The manuscript introduces extracellular vesicles, tetraspanins, and PD-L1 but does not sufficiently discuss the existing research gaps that the study aims to fill. For example:
1. What specific challenges or limitations exist in characterizing the sEV subpopulations in FF?
2. Why are the current methods inadequate for analyzing PD-L1+ sEVs, and how does this study provide a solution?
Clearly explain the novelty of using advanced techniques like single-particle interferometric reflectance imaging sensing (SP-IRIS) to overcome methodological obstacles.
Expand on the known roles of PD-L1+ EVs in regulating T-cell responses in cancer and hypothesize their similar role in promoting reproductive immunotolerance.
Highlight supporting evidence of PD-L1 enrichment in FF sEVs and their link to ovarian immune homeostasis.
While the study implies the potential use of FF sEVs as biomarkers for fertility outcomes, this idea is not explicitly introduced as a motivating factor. The impact of non-invasive biomarker research in assisted reproductive technology (ART) could underpin a stronger rationale.
Reference: Gu Y, Zhou G, Zhang M et al. Bioengineered extracellular vesicles presenting PD-L1 and Gal-9 to ameliorate new-onset primary ovarian insufficiency (POI). Chemical Engineering Journal. 2025/05/15/ 2025; 512:162635. doi: https://doi.org/10.1016/j.cej.2025.162635
The importance of tetraspanins (CD9, CD63, CD81) in sEV characterization is not adequately addressed in the Introduction. A more detailed explanation of why these markers are significant for identifying distinct EV subpopulations would enhance the understanding of their role in sEV characterization.
The introduction fails to highlight the methodological advancements of SP-IRIS and atomic force microscopy (AFM) compared to traditional techniques such as nanoparticle tracking analysis (NTA). As a result, the significance of employing novel methods is not fully emphasized.
The Introduction does not clearly articulate the specific knowledge gaps this study addresses. It does not address the following points:
Why are the existing characterization methods for FF sEVs inadequate?
• What specific questions about PD-L1+ sEV populations remain unanswered?
• How is the current understanding of sEV heterogeneity in FF limited?
Add a paragraph explicitly stating, "Despite the recognized importance of sEVs in reproductive biology, current understanding of their heterogeneity and functional subpopulations in human follicular fluid remains limited. Specifically, the presence, characteristics, and potential immunoregulatory roles of PD-L1+ sEV subpopulations have not been systematically investigated using high-resolution single-particle analysis techniques."
The rationale for using SP-IRIS and single-particle analysis is not adequately explained. It does not address the following points:
Why are the traditional bulk analysis methods insufficient?
• How does SP-IRIS overcome the limitations of conventional techniques?
• Why is single-vesicle resolution crucial for this research question?
The transition from background information to the specific research objectives appears to be lacking in logical flow. This is evident in several areas.
No clear statement of "therefore, we hypothesized..."
• Missing explanation of why PD-L1+ populations specifically need investigation
• Insufficient justification for focusing on tetraspanin co-expression patterns
The potential clinical applications and translational relevance are not clearly articulated as the driving forces for the research. These are:
• How understanding PD-L1+ sEV populations could impact fertility treatments
• Potential biomarker applications for reproductive outcomes
• Therapeutic implications for immune modulation in ART

Figures & Tables
The figures and tables provided are clear and legible.

Experimental design

Materials and Methods

The methods used in the study are described in commendable technical detail. A clear division of the experimental steps is presented, including patient recruitment, sEV isolation and characterization, and the use of advanced analytical techniques such as SP-IRIS and atomic force microscopy (AFM). The methods are presented systematically, allowing the reader to follow the experimental design. However, there are areas where additional details or clarifications would enhance the clarity further.
• While SP-IRIS is well described, additional information about the sensitivity, data processing, or fluorescence quantification methods would clarify how particle counts and protein co-localization were calculated.
• Patient recruitment details often provide information on age and hormonal data, but they frequently lack critical information regarding inclusion/exclusion criteria, such as lifestyle factors or medical history. This may introduce variability into the sEV data.
• Further description is needed on the validation of antibodies used in fluorescence labeling to show specificity for tetraspanins and PD-L1.
• The study lacks information on the use of appropriate negative and positive controls, including isotype controls and unstained extracellular vesicles, to validate the antibody-based immunofluorescence results.
The methods include significant detail, but some gaps could pose challenges for exact replication of the experiments. These are:
• While details on qEV columns are given, specifics such as buffer formulations, flow rate, and handling to optimize EV yield and purity are not detailed.
• Precise parameters for AFM imaging (e.g., resolution, scanning speed, particle selection algorithms) are not provided, making exact reproducibility challenging.
• Sample handling after SEC and potential aggregation or degradation of EVs during storage are not addressed, though these can impact downstream analysis.
• Although the study refers to the publication of data via Zenodo, raw data formats, calibration protocols, or intermediary analysis files were not openly included, limiting replication or external validation.
The study's timeframe appears to be sufficient for a cross-sectional characterization of small extracellular vesicles (sEVs) and tetraspanin/PD-L1 profiles. However, there is no indication that patient-derived follicular fluid was collected at multiple phases in the ovulatory cycle, which could potentially affect sEV abundance and protein expression. Incorporating a longitudinal element, such as tracking sEV characteristics over different stages of the menstrual cycle or peri-fertilization periods, might have provided more depth to the findings of the study.
The manuscript provides limited information regarding the statistical analysis, leaving several open questions about the reliability and robustness of the findings.
Only FF from 10 patients was examined, and no justification was provided regarding whether this sample size ensures adequate statistical power. For meaningful biological interpretation of PD-L1 expression in sEVs, a higher n is typically required, especially given potential inter-patient variability.
Perform a power analysis to justify the sample size.
There is no description of the statistical tests or metrics used to compare PD-L1 expression across groups or account for intra-sample variation.
• Were mean particle counts compared using t-tests, ANOVA, or non-parametric tests?
• Were error metrics or confidence intervals reported? These details must be included to assess alignment with objectives.
It is unclear whether biological replicate samples (e.g., multiple follicles per patient) were analyzed or patient data randomized to minimize bias.
Include bootstrapping or resampling techniques to account for variability between samples.
The hormonal context of patients undergoing Assisted Reproductive Technology (ART) cycles is not sufficiently standardized. Fluctuations in estradiol and progesterone levels during these cycles may significantly impact the release of extracellular vehicles (EVs) and the expression of tetraspanins. Furthermore, there is a lack of analysis of follicular fluid (FF) based on factors such as patient age subgroups, follicle size, or the phase of ovarian stimulation.
The study has several limitations that impact its reliability and translational value. One major issue is the lack of clear experimental controls, including negative or isotype controls, to confirm the specific binding of fluorescence labels. This omission makes it difficult to verify the accuracy of the results. Furthermore, the study does not provide sufficient experimental replicates, such as technical repeats or double-blinding of sample analysis, to minimize bias and random error. This lack of replication limits the confidence that can be placed in the findings.
The study does quantify PD-L1 expression, but it does not validate the potential immunoregulatory role of PD-L1+ sEVs through functional assays, such as immunosuppression assays or T-cell activation studies. As a result, the findings have limited translational impact.
The results are clearly described, but they lack robust statistical analysis. The study could have benefited from a more detailed presentation of variances in data, including metrics such as effect sizes, p-values, or error bands.

Validity of the findings

Results

The results presented in the manuscript show a high degree of novelty due to the application of advanced single-particle characterization techniques, specifically SP-IRIS and AFM, to study small extracellular vesicles from human follicular fluid. The identification of distinct subpopulations of vesicles that express PD-L1 and varying levels of tetraspanins (CD63, CD81, CD9) provides new insights into the diversity of extracellular vesicles within the ovarian microenvironment. Although research on extracellular vesicles in reproductive biology is an ongoing area of study, the specific focus on PD-L1 as an immune modulator in vesicles derived from follicular fluid addresses a lesser-understood aspect of reproductive immunology.
However, the results have some limitations, which are:
The study confirms rather than discovers the immunosuppressive role of PD-L1+ sEVs, largely extrapolated from knowledge in cancer biology, without providing functional validation in the ovarian context.
• Variations in tetraspanin-coexpression (CD9 vs. CD63/CD81) are interesting but not groundbreaking, as these markers are already widely studied in EV research (Reference: Karimi N, Dalirfardouei R, Dias T, Lotvall J, Lasser C. Tetraspanins distinguish separate extracellular vesicle subpopulations in human serum and plasma - Contributions of platelet extracellular vesicles in plasma samples. J Extracell Vesicles. May 2022;11(5):e12213. doi:10.1002/jev2.12213).
• The lack of robust correlation with clinical parameters, such as pregnancy outcomes or immune markers, limits its translational impact, which could elevate its novelty.
The manuscript does make a meaningful contribution to the field of reproductive biology. However, the study’s contribution is exploratory rather than transformative due to:
• The lack of functional assays to demonstrate how PD-L1+ sEVs modulate immune responses or interact with specific immune cells in the ovarian follicular environment.
• Insufficient analysis of clinical data (e.g., correlation between PD-L1+ sEV levels and pregnancy outcomes) to translate these findings into actionable insights for ART.
The dataset comprises FF from only 10 patients, which raises concerns about statistical power and generalizability. The lack of replication across diverse patient groups complicates broader assertions about the significance of PD-L1. Suggestion: Include a larger cohort or replicate findings in multiple populations to enhance confidence in the data.
• Negative controls (e.g., isotype controls or vesicles without PD-L1 labelling) are not explicitly mentioned. This omission could affect the reliability of fluorescence labelling and antigen capture results. Suggestion: Include detailed validation controls to mitigate the possibility of non-specific binding during SP-IRIS.
• While the study quantifies PD-L1+ sEV proportions, functional experiments (e.g., immune assays using T cells exposed to these sEVs) are absent. This creates a gap between the data presented and the biological implications of PD-L1 expression. Suggestion: Perform in vitro functional studies to validate the role of PD-L1+ sEVs in immune suppression.
• The study uses cross-sectional sEV analyses (single collection timepoint), which may miss dynamic changes in sEV populations over the ovarian cycle. This could limit the biological interpretation of the results.

Discussion

The findings described in the Discussion section generally correlate with the results presented in the manuscript. While the findings correlate with the results and are relevant to both the study's objectives and the broader research context, there are a few areas where the discussion could be enhanced.
The findings suggest a hypothetical immune-regulatory role for PD-L1+ sEVs in the ovarian microenvironment, but this is not validated with functional assays. For example, testing the effect of PD-L1+ sEVs on T-cell activity or cytokine secretion would strengthen the relevance of the findings to immune modulation.
The Discussion should acknowledge this limitation and propose further experiments to validate the functional role of PD-L1+ sEVs.
The potential for PD-L1+ sEVs to act as biomarkers is mentioned but not fully developed. The Discussion could explore how these findings might be applied in ART, linking PD-L1 expression to pregnancy outcomes or pathological conditions like ovarian immune dysregulation.
Recommendation: Provide specific examples of how PD-L1+ sEV profiling could aid clinical decision-making or fertility treatment strategies.
While PD-L1 is well-studied in cancer biology, its role in reproductive biology remains underexplored. The discussion could benefit from further comparisons to analogous mechanisms in other immune-tolerant systems, such as placental immunity.
Recommendation: Include references that compare PD-L1's role in other contexts (e.g., maternal-fetal interface immunity) to solidify its broader relevance.

Conclusion

The conclusions presented in the manuscript generally align with the findings obtained in the study. The main messages about identifying and characterizing PD-L1+ sEV subpopulations in human follicular fluid, their potential roles in modulating the immune system, and their implications for reproductive immunology are directly supported by the experimental results. However, while the conclusions are largely consistent with the findings, some areas would benefit from further explanation, justification, or additional context to strengthen the study's implications.
The manuscript lacks direct evidence showing that these vesicles play an active role in maintaining immune balance. To confirm this, functional tests such as co-culturing PD-L1-positive vesicles with T cells are necessary. The conclusion should be revised to highlight the need for experimental evidence to support the idea that these vesicles have immunoregulatory functions. The study does not directly show how the PD-1/PD-L1 pathway is activated or how it affects the immune system in the follicular microenvironment. It is essential to note that this is a hypothetical extension that requires both biological and mechanistic validation. Rephrase the statement: "The presence of PD-L1 in FF sEVs suggests that they are involved in maintaining immune homeostasis within the ovarian microenvironment." to: "The presence of PD-L1 in FF sEVs indicates potential roles in modulating immune signaling within the ovarian microenvironment, warranting further investigation through functional studies."
Add a more detailed discussion of potential clinical relevance or translational directions.
Expand the section on unanswered questions to guide future research. Include comparisons to other examples of PD-L1’s role in immune privilege.

Reviewer 2 ·

Basic reporting

The language and structure of the manuscript are fine. Figures are provided in poor quality, precluding a full understanding of the data. In particular, the texts in the legends and on the axis of plots are not visible. References used in the main text are missing in the bibliography (e.g. Thery et al, 2018, Salehi et al, 2023, Johnson, 2023; 2019, Luu et al, 2020). No raw data supplied, supplementary documents do not have useful data, and the Zenodo link provided does not work.

Experimental design

The scope of work is fine. While key references are cited in the main text, some of them are not listed in the bibliography, which makes it harder to evaluate the prior work that supported this research. The work aims to address the knowledge gap in the molecular phenotype of extracellular vesicles (EVs) in follicular fluids. The authors apply two types of analyses on FF-derived EVs (ExoView/SP-IRIS and AFM). However, the rigor of the investigation can be improved. The way the data is analyzed, presented, and discussed is confusing. Many assumptions made by the authors are not described to the reader, adding to the confusion. The authors did not list all materials for the experiments, such as the antibody used for PD-L1. Include catalogue numbers and the manufacturer/vendors of all reagents used.

Validity of the findings

Multiple claims are made, but the figures provided to support these claims are confusing or do not provide sufficient information.

Claim 1: “Our data indicate that the bulk of tetraspanin-expressing EVs in human FF are less than 50 nm in size.”
• Did not provide statistical analysis to support this claim. I believe the fluorescence mode includes both large (>=50 nm) and small (<50nm) particles. That means there should be a way to compare the number of tetraspanin-expressing large versus small particles with statistical analysis to support this claim.

Claim 2: “Tetraspanin and PD-L1 exhibit distinct expression and colocalization profiles at sEV level across all cohort samples.”
• It sounds like you are making at least two claims here, but they were not supported with statistical analysis.
• The first claim seems to be that PD-L1 and different tetraspanins colocalize to a different degree. Statistical analysis should be conducted in Figure 2B to support this claim.
• The second claim seems to be that this finding is consistent across the cohort. Again, statistical analysis should be conducted to show that there is no difference, regardless of which donor sample is analyzed.

Claim 3: “A total of 42%, 46%, and 50% of all the particles captured by anti-CD63, anti-CD81, and anti-CD9 antibodies, respectively, were positive for CD81.”
• Percentages are very confusing here. It begs the question: Shouldn’t all EVs captured by anti-CD81 be positive for CD81? I am sure there is a good explanation (which was not provided by the author), but reporting it as percentages is distracting from the point.

Claim 4: “PD-L1 was expressed at the highest level on CD9+ sEVs, with an average value of 5% within the cohort.”
• Like claim 2, you need to do a statistical analysis to make this claim.

Additional comments

In general, the authors did not provide information about how their plots are made. Are the bar plots mean +/- standard deviation or standard error of means? The description of the box plot is missing (where does the whisker extend to?) What are the sample numbers for each plot? Are they technical replicates (from multiple capture spots)? Did you pool the 10 clinical samples into one sample and analyze multiple times? Did you measure each specimen separately? Authors should show individual data points on the plots. Claims of distinct expression/co-localization or similarities between donors in the cohort must be supported by appropriate statistical analyses.

The discussion talks about non-invasive analysis, but collecting FF appears to be an invasive process.

Although the SP-IRIS platform can support unpurified samples, the current form of the analysis introduces bias because it requires that EVs be captured on the surface via tetraspanin antibodies.

Figure 1
• (A) Due to the low quality of the figure, the axis and legend for Figure 1 cannot be interpreted. Again, unclear what is being shown. According to the caption, this is a “Representative example”. Does that mean this is data from one of the donors, and it is showing technical replicates of one sample? For SP-IRIS (IM), there is an expectation of showing the size distribution, but this was not provided.
• (B & C) How were average particle counts calculated? Did you add up the total particles for each capture spot? Are you averaging across individual specimens from the cohort or technical replicates of capture spots or both? What is the error bar describing? Show individual data points on the plot.
• (D) Expressing as a percentage is very confusing. For example, for anti-CD81 detection, what does it mean when you report >100%? This means particles are likely to be double-counted, but this is distracting from the point you are making. The most straightforward way may be to report counts and make 3 separate bar plots for each capture antibody. Authors should look up previous work using SP-IRIS to gain insight into how data should best be presented.

Figure 2:
• (A &B) Again, expressing as percentages is very confusing. What is the total number of particles used to make these pie graphs? This data suggests that particles analyzed by IM (SP-IRIS) and fluorescence mode are completely distinct particles with no overlap. Is that true? Since the particles detected by IM were also captured by tetraspanin antibodies, I am certain that they would also be detected by tetraspanin fluorescent antibodies in fluorescence mode. Does that mean some particles are double-counted in the pie chart? Again, unsure what the error bars are, and whether we are looking at averages from individual specimens or technical replicates from a representative analysis of one donor. I think it will be very important here to show individual donor data in all plots because you are claiming that the finding is consistent across donors.

Figure 3
• Are those whiskers in the box plot extending to 1.5 interquartile range?
• How many particles are analyzed in each sample? Describe the number of AFM views analyzed for each sample
• Statistical analysis between the 3 samples?
• One flaw is that the samples analyzed by SP-IRIS and AFM were prepared in different ways (unpurified versus purified). So it is unknown whether this difference affected the sizing results. One analysis you can try is to compare the # of EVs above and below the 50 nm cutoff for both techniques to see if there is consistency in the %.

---

## Round 0.2 · accepted · Accept

All issues indicated by the reviewers were adequately addressed. The revised manuscript is acceptable now.

Reviewer 1 ·

Basic reporting

Title & Abstract
The authors have adequately addressed the comments. The authors have modified the title and added the details in the abstract as suggested.

Introduction
The authors have discussed the impact of FF sEV on fertility outcomes from the references, as suggested, and have also incorporated the details on tetraspanins. The authors have adequately addressed the comments and incorporated the research gaps for the study.

Figures & Tables
The tables and figures are legible and are adequate.

Experimental design

Material and Methods
The authors have adequately addressed the comments. The methodology now looks good with the suggested incorporations, and this is expected to enhance the reproducibility of the research.

Validity of the findings

Results
The authors have adequately responded to the queries and have incorporated the suggested changes. The authors have clearly mentioned the limitations of the study.

Discussion
The authors have made the appropriate changes in the manuscript, which have significantly improved the quality of the narrative. The addition of the limitations justified the pilot study and is adequate.

Conclusion
The changes incorporated by the authors are adequate and have improved the conclusion section of the manuscript.

Reviewer 2 ·

Basic reporting

Points have been addressed.

Experimental design

Points have been addressed.

Validity of the findings

Points have been addressed.

Additional comments

Points have been addressed.